# Screening of an Epigenetic Drug Library Identifies 4-((hydroxyamino)carbonyl)-*N*-(2-hydroxyethyl)-*N*-Phenyl-Benzeneacetamide that Reduces Melanin Synthesis by Inhibiting Tyrosinase Activity Independently of Epigenetic Mechanisms

**DOI:** 10.3390/ijms21134589

**Published:** 2020-06-28

**Authors:** Hyerim Song, Yun Jeong Hwang, Jae Won Ha, Yong Chool Boo

**Affiliations:** 1Department of Molecular Medicine, School of Medicine, Kyungpook National University, Daegu 41944, Korea; happylove951@naver.com (H.S.); hwangkyy26@naver.com (Y.J.H.); jaewon1226@knu.ac.kr (J.W.H.); 2Brain Korea (BK) 21 Plus Kyungpook National University (KNU) Biomedical Convergence Program, Kyungpook National University, Daegu 41944, Korea; 3Cell and Matrix Research Institute, Kyungpook National University, Daegu 41944, Korea

**Keywords:** melanin, epigenetic screening library, antimelanogenic drug, tyrosinase, pigmentation, 4-((hydroxyamino)carbonyl)-*N*-(2-hydroxyethyl)-*N*-phenyl-benzeneacetamide, HPOB, tyrosinase inhibitor

## Abstract

The aim of this study was to identify novel antimelanogenic drugs from an epigenetic screening library containing various modulators targeting DNA methyltransferases, histone deacetylases, and other related enzymes/proteins. Of 141 drugs tested, K8 (4-((hydroxyamino)carbonyl)-*N*-(2-hydroxyethyl)-*N*-phenyl-benzeneacetamide; HPOB) was found to effectively inhibit the α-melanocyte-stimulating hormone (α-MSH)-induced melanin synthesis in B16-F10 murine melanoma cells without accompanying cytotoxicity. Additional experiments showed that K8 did not significantly reduce the mRNA and protein level of tyrosinase (TYR) or microphthalmia-associated transcription factor (MITF) in cells, but it potently inhibited the catalytic activity TYR in vitro (IC_50_, 1.1–1.5 µM) as compared to β-arbutin (IC_50_, 500–700 µM) or kojic acid (IC_50_, 63 µM). K8 showed copper chelating activity similar to kojic acid. Therefore, these data suggest that K8 inhibits cellular melanin synthesis not by downregulation of TYR protein expression through an epigenetic mechanism, but by direct inhibition of TYR catalytic activity through copper chelation. Metal chelating activity of K8 is not surprising because it is known to inhibit histone deacetylase (HDAC) 6 through zinc chelation. This study identified K8 as a potent inhibitor of cellular melanin synthesis, which may be useful for the treatment of hyperpigmentation disorders

## 1. Introduction

Human skin color is controlled by the composition and distribution of various chromophores such as melanin, hemoglobin, and carotenoids [1]. Melanin is a dark polymer pigment synthesized by melanocytes [2]. Melanosomes containing melanin migrate from melanocytes to neighboring keratinocytes via dendrites, resulting in the distribution of melanin throughout the epidermis [3]. Melanin plays an important role in protection against carcinogenic ultraviolet radiation [4]; therefore, melanin metabolism is critical in epidermal homeostasis [5,6].

Dysfunctions associated with melanin synthesis cause clinically relevant pigmentation disorders that can be congenital or acquired, cutaneous or systemic, temporary or permanent, and related to hypo- or hyperpigmentation [7,8]. Hyperpigmentation disorders represent skin conditions in which dark pigment is deposited abnormally, unevenly, or excessively either via intrinsic pathophysiological factors, or extrinsic environmental factors [9]. Hyperpigmentation can occur after an acute inflammatory reaction induced by pathological agents and physicochemical damage, or as a natural process of skin aging [10].

Genetic skin color differences are determined by genetic traits such as polymorphism in solute carrier family 24 member 5 (SLC24A5) and solute carrier family 24 member 5 (SLC24A5) genes [11,12]. Single nucleotide polymorphisms of these genes can alter the activity of potassium-dependent sodium-calcium exchangers and affect melanosome biosynthesis and melanin synthesis [13,14]. 

In addition to genetic traits, epigenetic mechanisms, which include DNA methylation, histone modifications, and microRNAs, can also mediate long-term phenotypic changes in skin cells [15]. A DNA methylation inhibitor, 5-azacytidine, was shown to decrease melanin levels in Mel-Ab cells by downregulating microphthalmia-associated transcription factor (MITF) and its regulatory target tyrosinase (TYR, EC 1.14.18.1), via inactivation of cAMP response element-binding protein (CREB) [16]. Histone deacetylase (HDAC) inhibitor drugs, including trichostatin A, were found to repress melanin synthesis through transcriptional downregulation of the melanocyte-specific isoform of MITF (M-MITF) in melanocytes, melanoma, and clear cell sarcoma cells, and topical application of trichostatin A suppressed skin pigmentation in mice [17]. Among microRNAs, miR-125b was found to decrease melanin levels and it showed reduced expression in pigmented cells because of promoter hypermethylation [18]. The target of miR-125b was identified as SRC homology 3 domain-binding protein 4 (SH3BP4), the transcriptional expression of which is controlled by MITF [19].

TYR is a copper-containing enzyme that catalyzes the hydroxylation of L-tyrosine to L-3,4-dihydroxyphenylalanine (DOPA), and the oxidation of L-DOPA to its corresponding o-quinone derivative in the melanin synthetic pathway [20]. A variety of natural and synthetic compounds are known to inhibit the catalytic activity of TYR by multiple mechanisms [21,22]. For example, β-arbutin and kojic acid are known to inhibit the catalytic activity of TYR in a competitive manner TYR [23,24].

The present study was undertaken to screen an epigenetic drug library to identify new drug candidates for pharmacotherapy of hyperpigmentation. Of 141 total drugs tested in this study, drugs H4 (*N*-hydroxy-3-C(3-(hydroxyamino)-3-oxo-1-propen-1-yl)-benzamide; CBHA) and K8 (4-((hydroxyamino)carbonyl)-*N*-(2-hydroxyethyl)-*N*-phenyl-benzeneacetamide; HPOB) were found to inhibit α-melanocyte-stimulating hormone (α-MSH)-induced melanogenesis in B16-F10 murine melanoma cells at 1.0 μM without significant cytotoxicity. Of these two compounds, K8 showed a superior activity to the toxicity profile and was chosen for further mechanistic study.

K8 is known to inhibit HADC 6 through zinc chelating activity [25]. We initially expected K8 to be able to reduce melanin by inhibiting TYR expression by epigenetic mechanisms, but unexpectedly, no strong evidence for this action could be found. On the other hand, K8 was shown to strongly inhibit the catalytic activity of TYR in a competitive manner, probably through copper chelating activity. Here we report that screening of an epigenetic drug library resulted in the identification of new drug candidates for the pharmacotherapy of skin hyperpigmentation disorders.

## 2. Results

### 2.1. Screening of a Drug Library for Antimelanogenic Activity

A commercially available epigenetic screening library was used in the present study. The composition of the library is shown in Table 1. The library contains 141 cell-permeable small molecule drugs that modulate the activity of epigenetic enzymes including DNA methyltransferases, DNA demethylases, histone acetyltransferases, HDACs, or acetylated histone binding proteins. 

In the preliminary screening assay, each drug was tested for its effects on melanin synthesis in murine melanoma-derived B16-F10 cells cultured in vitro. After treating the cells with a drug solution (final concentration, 1.0 μM) or vehicle, melanin synthesis was stimulated with α-MSH (final concentration, 100 nM). As shown in Figure 1a, α-MSH increased cellular melanin synthesis, as monitored by optical density (OD) at 400 nm, and the induction was modulated by the various drugs. The effects of the drugs on cell viability were also determined as shown in Figure 1b. Comparing activity and cytotoxicity, drugs H4 and K8 were selected for additional study because they markedly inhibited cellular melanin synthesis without reducing cell viability at the tested concentration. 

### 2.2. Activity Versus Cytotoxicity for H4 and K8

These two drugs, H4 (CBHA) and K8 (HPOB) were compared in the next experiment for their antimelanogenic activity and cytotoxicity. B16-F10 cells were treated with drug at varied concentrations and stimulated with 100 nM α-MSH for 72 h. As shown in Figure 2a,b, the increase in OD 400 due to melanin synthesis was attenuated by both drugs in a dose-dependent manner. However, drug H4 showed significant cytotoxicity above 3 μM, whereas K8 did not show toxic effects up to 10 μM (Figure 2c,d). Thus, K8 was selected for further mechanistic study. 

### 2.3. Effects of K8 on Melanin Content and TYR Activity in Cells

Additional assays were undertaken to separately quantify extracellular and intracellular melanin levels of cells treated with K8. As shown in Figure 3a,b, both the extracellular and intracellular melanin content, normalized for total cellular protein content, were increased by α-MSH treatment, which was suppressed by K8 in a dose-dependent manner (1–10 µM). 

Changes in melanin content of cells may be closely related to TYR activity in cells. In order to directly examine this possibility, cellular TYR activity was assessed after treating different concentrations of drugs and a fixed amount of α-MSH. As a result, α-MSH increased cellular TYR activity, and the change was attenuated by K8 (Figure 3c). The less expected observation was that the TYR activity of the cells was only partially lowered by 10 µM K8 that suppressed almost all melanin synthesis as observed above.

### 2.4. Effects of K8 on the mRNA and the Protein Expression Levels of TYR, TRP1, and DCT

We further examined the effects of K8 on the gene expression of TYR and other related enzymes involved in melanogenesis. Expression levels of the mRNAs for TYR, tyrosinase-related protein 1 (TYRP1), and dopachrome tautomerase (DCT) were assessed in B16-F10 cells by quantitative reverse transcription polymerase chain reaction (qRT-PCR) using to glyceraldehyde 3-phosphate dehydrogenase (GAPDH) as a control (Figure 4a–c). α-MSH significantly increased the mRNA levels of TYR and TRP1, whereas the change in DCT was insignificant. mRNA levels of TYR, TYRP1, and DCT were not significantly affected by K8. 

Western blotting showed that α-MSH also increased the protein levels of TYR, TYRP1, and DCT in B16-F10 cells (Figure 4d–f). K8 did not significantly attenuate the increase in the protein levels of TYR, TRP1, and DCT caused by α-MSH stimulation.

### 2.5. Effects of K8 on the mRNA and the Protein Expression Levels of MITF

MITF is a key transcription factor that regulates the gene expression of TYR and other melanogenic enzymes [2,8]. As shown in Figure 5a,b, α-MSH induced MITF mRNA and protein expression but these changes were not significantly affected by K8. 

### 2.6. Effects of K8 on the Catalytic Activity of Murine and Human TYRs

Despite the potent inhibitory effects of K8 on cellular melanin synthesis, the mRNA and protein levels of TYR and other related enzymes were not significantly inhibited. This suggests that mechanisms other than gene expression regulation exist. As one possibility, we examined whether the drug directly inhibits the catalysis of TYR. Two different TYR sources were used for this purpose. One is the protein extract of B16F10 cells stimulated by α-MSH and thus expresses a high amount of TYR. The other is the protein extract of HEK293-TYR, a stable cell line overexpressing human TYR. In these experiments, the effect of K8 on the catalytic activity of TYR was compared with that of β-arbutin or kojic acid [23]. As shown in Figure 6a, K8 appeared to inhibit murine TYR (IC_50_, 1.5 µM) much more potently than β-arbutin (IC_50_, 700 µM). As shown in Figure 6b, K8 (IC_50_, 1.1 µM) inhibited and TYR much more potently than β-arbutin (IC_50_, 500 µM) and kojic acid (IC_50_, 63 µM).

### 2.7. Enzyme Kinetic Study to Determine Inhibition Types of K8 and β-arbutin.

As shown in Figure 7a, the lines of Lineweaver–Burk plots for TYR activity obtained from the uninhibited enzyme and different concentrations of K8 virtually converged to the same intercept on the 1/activity axis, indicating a competitive inhibition. As shown in Figure 7b, the study with varying concentrations of β-arbutin also resulted in a group of lines with the same intercept on the 1/activity axis, indicating a competitive inhibition. 

### 2.8. Copper Chelating Activity of K8 and Kojic Acid

Pyrocatechol violet was used to compare the copper chelating activities of K8, kojic acid, and β-arbutin. As shown in Figure 8a, the absorption spectrum of pyrocatechol changed in the presence of copper ions. The increase of absorbance of pyrocatechol violet at 632 nm due to copper chelation was attenuated by K8 and kojic acid, but not by β-arbutin (Figure 8b). Equimolar concentrations (200 μM) of K8 and kojic acid were estimated to chelate 27% and 36% of 200 μM copper ions, respectively, whereas β-arbutin did not show copper chelating activity.

### 2.9. Screening of a Drug Library for TYR Inhibitory Activity

The next experiment examined the library for other TYR inhibitors. As a result of evaluating the human TYR inhibitory effect of each of 141 compounds from the library at 1.0 μM, it was found that not only K8 and H4, but A5 (PCI 34051; *N*-Hydroxy-1-((4-methoxyphenyl)methyl)-1*H*-indole-6-carboxamide), C10 (ITF 2357; givinostat; (6-((diethylamino)methyl)naphthalen-2-yl)methyl (4-(hydroxycarbamoyl)phenyl)carbamate hydrochloride hydrate), and M3 (MC 1568; 3-(5-(3-(3-Fluorophenyl)-3-oxoprope n-1-yl)-1-methyl-1H-pyrrol-2-yl)-*N*-hydroxy-2-propenamide) also had potent TYR inhibitory effects (Figure 9). 

## 3. Discussion

Melanin synthesis is an important research topic in the context of skin pigmentation disorders [6,26,27]. Skin pigmentation disorders can cause severe mental stress that reduces productivity and quality of life [28] and are thus considered a socially important disease. Various approaches are available for the treatment of hyperpigmentation, such as chemical peeling, laser surgery, pharmacotherapy, and cosmetic camouflage [29,30,31], but these remain unsatisfactory. Hydroquinone is a commonly used treatment for skin pigmentation disorders that can be administered alone or in combination with other drugs, but it is associated with side effects such as irritation, allergy, and cancer [32]. Although various inhibitors of melanin synthesis are used in the cosmetic industry to control skin pigmentation, consumer satisfaction is low, and more effective skin lightening treatments are in high demand [33,34,35]. 

Melanin is synthesized in cells through a multistep reaction that begins with the oxidation of L-tyrosine and/or L-DOPA to L-dopaquinone catalyzed by the TYR enzyme [36]. Thus, TYR is a useful target for the control of unwanted skin pigmentation, and several strategies have been proposed to target TYR for the control melanin synthesis [21,37]. Gene expression of melanogenic enzymes, such as TYR, TYRP1, and DCT, is induced by MITF in melanocytes [2,8]. Melanocortin 1 receptor (MC1R) is stimulated by α-MSH and other proopiomelanocortin-derived peptide hormones, which activates adenylate cyclase [8,38]. The increase in cyclic AMP subsequently activates protein kinase A and induces MITF gene expression [39]. MITF gene expression is also induced by other mechanisms involving the c-kit or WNT pathways [40].

Our team has been exploring natural products and peptides to find more effective melanin inhibitors. As a result, it was shown that the extract of *Sasa quelpaertensis* effectively inhibited the production of melanin and its main component, p-coumaric acid, inhibited the activity of the TYR enzyme in a manner competing with the substrate [41,42,43]. In addition, in an exploratory study to discover human TYR inhibitors from terrestrial plant extracts, it was observed that the extract of *Vitis viniferae* and its active ingredient, resveratrol, strongly inhibited human TYR and thereby inhibited cell melanin production [44,45,46]. In a study of marine plants, it was reported that the extract of *Phyllospadix iwatensis* and its active ingredient, luteolin 7-sulfate, inhibited cell melanin production by inhibition of TYR gene expression as well as inhibition of TYR catalytic activity [47,48]. Recently, by applying the peptide library screening method, we found low molecular peptides that inhibit cell melanin production [49,50]. These peptides share the common sequence with α-MSH and thus can act as an antagonist of MC1R which are involved in the regulation of TYR gene expression [50]. 

The current study was carried out on an extension of the research to find an effective and safe melanogenesis modulator. This study is different from the previous studies in that an epigenetic drug library was used. The aim at the beginning of this study was to find a substance that modulates TYR expression by epigenetic mechanisms. However, K8, a drug selected to safely and strongly inhibit melanin production through cell-based assay, did not regulate TYR gene expression, but instead directly regulated the catalytic activity of TYR. Our original assumption was that K8 might inhibit melanin synthesis by suppressing TYR expression to the same extent (as monitored by Western blotting and enzyme assay using the cell lysates) but it turned out not to be the case. As a result, although a drug with a mechanism different from the initial hypothesis was found, this study was considered very meaningful and successful because it provided a new drug candidate with very good potency. K8 was originally developed as an HDAC 6-selective inhibitor to enhance the effectiveness of DNA-damaging anticancer drugs [25]. However, its antimelanogenic effect had not been reported prior to this study. 

In the first cell-based screening assay using the drug library, the melanin synthesis inhibitory effect of 141 drugs was compared at 1.0 μM. As a result, two drugs, H4 and K8, were selected because of their strong melanin synthesis inhibitory effect without cytotoxicity. Of these, because H4 showed toxicity at 3 μM and higher concentrations and K8 was finally elected as a promising drug candidate. 

In the final in vitro screening assay using the drug library, human TYR inhibitory activity of 141 drugs was compared at 1.0 μM. In addition to H4 and K8, three more drugs, A5, C10, and M3, showed excellent inhibitory action. Of these, A5 and M3 showed no melanin inhibitory action and C3 showed strong cytotoxicity in the cell-base experiments. Therefore, among the drugs library, K8 was selected as the best new drug candidate. 

The drugs that had shown cytotoxicity at 1 μM (Figure 1) were eliminated from the active drug candidates. The eliminated drugs might have inhibited melanin synthesis without cytotoxicity if they had been assayed at lower concentrations. Therefore, further studies are needed to examine this possibility. Because only five out of 141 drugs inhibited the catalytic activity of TYR (Figure 1), additional active drugs, if found any, are expected to inhibit melanin synthesis by mechanisms other than inhibition of TYR catalytic activity.

We also examined whether K8 can reduce the gene expression levels of TYR and other related enzymes involved in melanin synthesis via the regulation of transcription factors such as MITF. The results showed that K8 did not significantly reduce the mRNA and protein levels of TYR, TYRP1, DCT, or MITF. Therefore, the inhibitory effect of the drug on cellular melanogenesis could not be explained by the reduced gene expression of TYR or other related enzymes/proteins. We then investigated other potential mechanisms for the antimelanogenic effects of K8 and found that this drug could directly inhibit the catalytic activity of murine TYR (IC_50_, 1.5 µM) much more potently than β-arbutin (IC_50_, 700 µM). Like β-arbutin, K8 showed a competitive type of inhibition against TYR activity, as determined by Lineweaver–Burk plots. 

Our team previously developed an HEK293-TYR cell line that constitutively expresses human TYR and used it in the screening assay to identify potent human TYR inhibitors [51,52]. As a result, several compounds were found to potently inhibit human TYR, including *p*-coumaric acid (IC_50_, 4.0 µM), resveratrol (IC_50_, 1.7 µM), and luteolin 7-sulfate (IC_50_, 8.2 µM) [43,45,47]. These compounds have been shown to exhibit antimelanogenic effects via multiple mechanisms [43,46,48]. The IC_50_ value of K8 for human TYR was estimated at 1.1 µM, which is much lower than that of β-arbutin (500 µM) or kojic acid (IC_50_, 63 µM). It is also a more potent TYR inhibitor compared with the aforementioned *p*-coumaric acid, resveratrol, and luteolin 7-sulfate. 

Kojic acid can competitively inhibit TYR activity through a mechanism involving copper cheating activity [24]. In addition, numerous hydroxamic acid derivatives can chelate metal ions and inhibit metalloenzymes such as TYRs and HDACs [53,54]. K8 is a hydroxamic acid derivative and is known to inhibit the activity of HDAC 6 through the chelation of zinc ions [25]. In the present study, K8 and kojic acid were observed to have similar copper chelating activity and β-arbutin has no such activity. Therefore, the mechanism of action of K8 seems to be more similar to kojic acid than β-arbutin. 

Baek et al. compared the melanin production inhibitory effect and mushroom TYR inhibitory activity of kojic acid and several hydroxamic acid derivatives [55]. They observed that 4-(Adamantanecarboxamido)-*N*-hydroxybenzamide (1a) and N1-Adamantyl-N4-hydroxyterephthal amide (1b) inhibited cellular melanin production (IC_50_, 3.78 µM and 7.46 µM, respectively) more potently than kojic acid (IC_50_, 2.0 mM). They also inhibited the catalytic activity of mushroom TYR in vitro (1a, IC_50_, 12.04 µM; 1b, IC_50_, 8.99 µM) more potently than kojic acid (IC_50_, 50.20 µM). Compared to these two hydroxamic acid derivatives as well as kojic acid indirectly, K8 was estimated to have more potent inhibitory activity against TYR catalytic activity. 

The skin pigmentation degree will be determined by the relative balance between the pigmentation processes, such as melanin synthesis, and the depigmentation processes, such as the elimination of stratum corneum. We have previously reported that topically applied resveratrol derivatives, which inhibited melanin synthesis through multiple mechanisms including inhibition of TYR catalytic activity [44,46,56], showed depigmentation efficacy in human trials [34,35]. Thus, if the depigmentation process is in the same condition, the inhibition of melanin synthesis may brighten the skin tone, and a potent TYR inhibitor may be a candidate drug. Further research is needed to investigate whether the active drug found in this study can be delivered to melanocytes well and whether it can actually exhibit pigmentation efficacy in vivo.

## 4. Materials and Methods

### 4.1. Reagents and a Drug Library

An epigenetic screening library (Item No. 11076) was purchased from Cayman Chemical (Ann Arbor, MI, USA). It contains 141 cell-permeable small molecules that modulate the activity of epigenetic enzymes, such as DNA methyltransferases, DNA demethylases, histone acetyltransferases, HDACs, or acetylated histone binding proteins. The composition of the library is shown in Table 1. Drugs H4 (*N*-hydroxy-3-[3-(hydroxyamino)-3-oxo-1-propen-1-yl-benzamide; CBHA) and K8 (4-((hydroxyamino)carbonyl)-*N*-(2-hydroxyethyl)-*N*-phenyl-benzeneacetamide; HPOB) were purchased from Cayman Chemical. α-MSH (M4135) and β-arbutin (4-hydroxyphenyl-β-D-glucopyranoside, A4256) were purchased from Sigma-Aldrich (St. Louis, MO, USA).

### 4.2. Cell Culture

Cells were cultured at 37 °C in a humidified atmosphere of 5% CO_2_ and 95% air. B16-F10 murine melanoma cells (ATCC CRL-6475) obtained from the American Type Culture Collection (Manassas, VA, USA) were cultured in Dulbecco’s modified Eagle’s medium (DMEM) containing 10% fetal bovine serum (FBS, Gibco BRL, Grand Island, NY, USA) and antibiotics (100 U∙mL^−1^ penicillin, 0.1 mg∙mL^−1^ streptomycin, 0.25 μg∙mL^−1^ amphotericin B, Thermo Fisher, Waltham, MA, USA). 

### 4.3. Screening Assay for Overall Melanin Synthesis

For the screening assay, B16-F10 cells were plated in 96-well plates (3 × 10^3^ cells per well) and cultured for 24 h. The cells were then treated with a test drug at 1.0 μM for 60 min, and subsequently stimulated with 0.1 μM α-MSH for 72 h. Overall melanin synthesis was estimated by measuring the absorbance of each well (cells plus medium, without cell lysis) at 400 nm using a Spectrostar Nano microplate reader (BMG LABTECH GmbH, Ortenberg, Germany).

### 4.4. Cell Viability Assay

Cell viability was evaluated using 3-(4,5-dimethylthiazol-2-yl)-2,5-diphenyltetrazolium bromide (MTT) [26]. After treatment of B16-F10 cells with each drug for 48 h, cells were washed with phosphate-buffered saline (PBS) and incubated for 2 h in the culture medium containing 1 mg∙mL^−1^ MTT (Amresco, Solon, OH, USA). The medium was discarded by aspiration, the formazan dye deposited inside the cells was extracted with isopropanol, and absorbance was measured at 595 nm using a Spectrostar Nano microplate reader.

### 4.5. Melanin Content Assay

The intracellular melanin retained in cells and extracellular melanin secreted from cells were separately determined [49,57]. B16-F10 cells were plated in 6-well plates (1.2 × 10^4^ cells per well), cultured for 24 h, treated with a test drug at various concentrations for 60 min, and stimulated with 100 nM α-MSH for 72 h. The conditioned medium was evaluated to determine the extracellular melanin content. Adherent cells were extracted with 10% dimethyl sulfoxide solution containing 1.0 M sodium hydroxide at 60 °C for 60 min, and melanin content in the extract was estimated by measuring the absorbance at 400 nm and normalized to the total protein content of the cells.

### 4.6. Cellular TYR Activity Assay

After B16-F10 cells were treated with drug and/or α-MSH, the TYR activity of cell extracts was determined by an indirect spectrophotometric method to monitor dopachrome formation with L-tyrosine plus L-DOPA as the substrates [48,58]. Total cellular protein was extracted from the treated cells using a lysis buffer of 10 mM Tris-HCl buffer (pH 7.4) containing 120 mM NaCl, 25 mM KCl, 2.0 mM EDTA, 1.0 mM EGTA, 0.5% Triton X-100, and a protease inhibitor cocktail (Roche, Mannheim, Germany) at 4 °C, followed by centrifugation at 13,000× *g* for 15 min. The TYR activity assay mixture (200 μL) consisting of 100 mM sodium phosphate buffer (pH 6.8), 1.0 mM L-tyrosine, and 42 μΜ L-DOPA was incubated with 40 μg cellular protein extract at 37 °C, and changes in absorbance at 475 nm over time were measured using a Spectrostar Nano microplate reader. 

### 4.7. qRT-PCR Analysis

After B16-F10 cells were treated with drug and/or α-MSH, cellular RNA was extracted using an RNeasy kit (Qiagen, Valencia, CA, USA) and the complementary DNA (cDNA) was synthesized by reverse transcription using a High-Capacity cDNA Archive Kit (Applied Biosystems, Foster City, CA, USA). qRT-PCR was conducted using a StepOnePlus™ Real-Time PCR System (Applied Biosystems). The reaction mixture (20 μL) comprised SYBR^®^ Green PCR Master Mix (Applied Biosystems), 60 ng cDNA, and 2 pmol gene-specific primer sets (Macrogen, Seoul, Korea). The sequences of the primers used in this study are listed in Table 2. 

The reaction protocol was as follows: initial incubation at 50 °C for 2 min; DNA polymerase activation at 95 °C for 15 min; and annealing and extension at 60 °C for 1 min. In each PCR run, the melting curve was checked to confirm the homogeneity of the PCR product. The mRNA expression levels of the gene of interest were determined using the comparative Ct method, and normalized to the internal reference *GAPDH* [59]. In this method, Ct is defined as the number of PCR cycles required for the signal to exceed the threshold. Fold changes in the test group compared to the control group were calculated as 2^−ΔΔCt^, where ΔΔCt = ΔCt_(test)_ − ΔCt_(control)_ = [Ct_(gene, test)_ − Ct_(reference, test)_] − [Ct_(gene, control)_ − Ct_(reference, control)_]. 

**Table 2 ijms-21-04589-t002:** Sequences of primers used for quantitative reverse transcriptase polymerase chain reaction.

Gene Name	GenBank Accession Number	Primer Sequences	References
*TYR*	NM_011661.5	Forward: 5′-CTTCTTCTCCTCCTGGCAGATC-3′ Reverse: 5′-TGGGGGTTTTGGCTTTGTC-3′	[60]
*TYRP1*	NM_001282015.1	Forward: 5′-CAGTGCAGCGTCTTCCTGAG-3′ Reverse: 5′-TTCCCGTGGGAGCACTGTAA-3′	[61]
*DCT*	NM_010024.3	Forward: 5′-GCAAGAGATACACGGAGGAAG-3′ Reverse: 5′-CTAAGGCATCATCATCATCACTAC-3′	[62]
*MITF*	NM_008601.3	Forward: 5′-GCTGGAAATGCTAGAATACAG-3′ Reverse: 5′-TTCCAGGCTGATGATGTCATC-3′	[60]
*GAPDH*	NM_001289726.1	Forward: 5′-GCATCTCCCTCACAATTTCCA-3′ Reverse: 5′-GTGCAGCGAACTTTATTGATGG-3′	[63]

### 4.8. Western Blotting

Western blotting was performed as previously described [49,64]. Primary antibodies for TYR (#127217) was purchased from MyBioSource (San Diego, CA, USA). Primary antibodies for TYRP1 (#10443) and β-actin (#47778) were purchased from Santa Cruz Biotechnology (Santa Cruz, CA, USA). Primary antibodies for MITF (#20663) and DCT (#74073) were purchased from Abcam (Cambridge, UK). Anti-rabbit IgG (#2357) and anti-goat IgG (#2020) secondary antibodies were purchased from Santa Cruz Biotechnology. Anti-mouse IgG (#7076) secondary antibody was purchased from Cell Signaling Technology (Danvers, MA, USA). Each antibody was diluted 1 to 1000 in antibody dilution buffer containing 20 mM Tris-Cl (pH 7.5), 200 mM NaCl and 5% skim milk.

After B16-F10 cells were treated with drug and/or α-MSH, whole-cell lysates were prepared using a lysis buffer (pH 7.2) consisting of 10 mM Tris-Cl (pH 7.4), 120 mM NaCl, 25 mM KCl, 2 mM EGTA, 1 mM EDTA, 0.5% Triton X-100 and a protease inhibitor cocktail (Roche, Mannheim, Germany) for foe evaluation of TYR, TYRP1, and DCT. For MITF extraction, cells were extracted with RIPA buffer consisting of 25 mM Tris-HCl (pH 7.6), 150 mM NaCl, 1% NP-40, 1% sodium deoxycholate, 0.1% SDS and protease inhibitor cocktail. Proteins were denatured in the Laemmli sample buffer by heating at 95 °C for 5 min and separated by 7.5% SDS-PAGE gel. Proteins were transferred to a polyvinylidene difluoride membrane (Amersham Pharmacia, Little Chalfont, UK). The membrane was incubated with a primary antibody solution at 4 °C overnight, followed by incubation with a solution of a secondary antibody conjugated to horseradish peroxidase at room temperature for 1 h. The target protein bands were detected by a chemiluminescence method using a picoEPD Western Reagent kit (ELPIS-Biotech, Daejeon, Korea). The blots were analyzed using the NIH Image J program.

### 4.9. In Vitro TYR Catalytic Activity Assay 

To examine the direct effect of a drug on the catalytic activity of TYR, in vitro experiments were undertaken. Lysate of B16F10 cells stimulated with α-MSH for 24 h was used as a murine TYR preparation. Human TYR preparation was the lysate of human embryonic kidney 293 cells constitutively expressing human TYR (HEK293-TYR) [51,52]. The cells were lysed in 10 mM Tris-HCl buffer (pH 7.4) containing 120 mM NaCl, 25 mM KCl, 2.0 mM EDTA, 1.0 mM EGTA, 0.5% Triton X-100, and a protease inhibitor cocktail (Roche) at 4 °C and then centrifuged at 13,000× *g* for 15 min to obtain the supernatants used as TYR preparations. The TYR activity assay mixture (200 μL) consisted of 100 mM sodium phosphate buffer (pH 6.8), TYR preparation (40 μg protein), a drug at various concentrations, 1.0 mM L-tyrosine, and 42 μΜ L-DOPA. For enzyme kinetic study, the concentrations of L-tyrosine plus L-DOPA were varied while their relative ratio was kept the same as the standard assay condition. The mixture was incubated at 37 °C, and absorbance at 475 nm was monitored using a Spectrostar Nano microplate reader. The reactions were also performed in the absence of a TYR preparation to correct for the effects of non-enzymatic reactions. Enzyme activity was calculated as follows: activity (% control) = (C−D)/(A−B) × 100, where A and B are the absorbance changes of the control group with and without enzyme, respectively, while C and D are the absorbance changes of the test group with and without enzyme, respectively. 

### 4.10. Copper Chelating Activity Assay

Copper chelating activity was determined by using pyrocatechol violet (Sigma-Aldrich) [24]. The principle of this method is that, when the pyrocatechol violet is complexed with copper, it shows maximum absorption at 632 nm, and the test material competing with pyrocatechol violet for copper can reduce the absorbance of the copper–pyrocatechol violet complex. The reaction mixture containing a potential chelator at varied concentrations, 200 μM CuSO_4_, and 200 μM pyrocatechol violet was incubated at 25 °C for 20 min, and its absorption spectra were recorded using a Shimadzu UV-1650PC spectrophotometer (Shimadzu Corporation, Kyoto, Japan). Copper chelating activity was evaluated by the change in the absorbance of pyrocatechol violet at 632 nm. 

### 4.11. Statistical Analyses

The data are expressed as the mean ± standard deviation (SD) of three independent experiments. The experimental results were statistically analyzed using SigmaStat v. 3.11 software (Systat Software Inc., San Jose, CA, USA), by one-way analysis of variance (ANOVA), followed by Dunnett’s test.

## 5. Conclusions

In conclusion, the present study identified K8 from an epigenetic drug library as a potent inhibitor of cellular melanin synthesis, suggesting its potential utility in the treatment of hyperpigmentation disorders. The results suggest that K8 inhibits cellular melanin synthesis not through “epigenetic” downregulation of TYR protein levels but by direct inhibition of TYR catalytic activity. Further studies are needed to examine its in vivo and clinical efficacy.

## Figures and Tables

**Figure 1 ijms-21-04589-f001:**
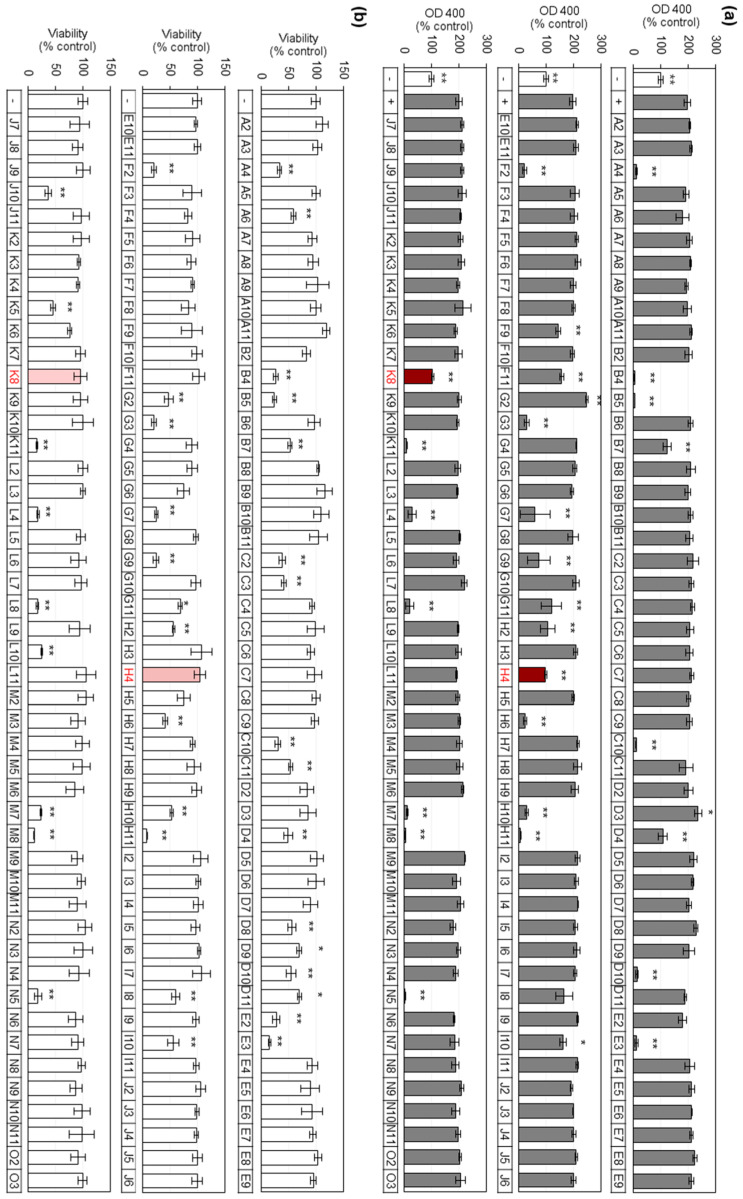
Effects of drugs on melanin synthesis and viability in B16-F10 cells. (**a**) Cells were treated with each drug at 1.0 μM and stimulated with 100 nM α-melanocyte-stimulating hormone (α-MSH) for 72 h for relative melanin synthesis assay (mean ± SD, *n* = 3). ** *p* < 0.01, * *p* < 0.05 versus α-MSH control. (**b**) Cells were treated with each drug at 1.0 μM for 48 h for relative viability assay (mean ± SD, *n* = 3). ** *p* < 0.01, * *p* < 0.05 versus control. Open and closed histograms represent the absence and presence of α-MSH addition, respectively. Drugs that inhibited melanin synthesis without cytotoxicity are highlighted in red.

**Figure 2 ijms-21-04589-f002:**
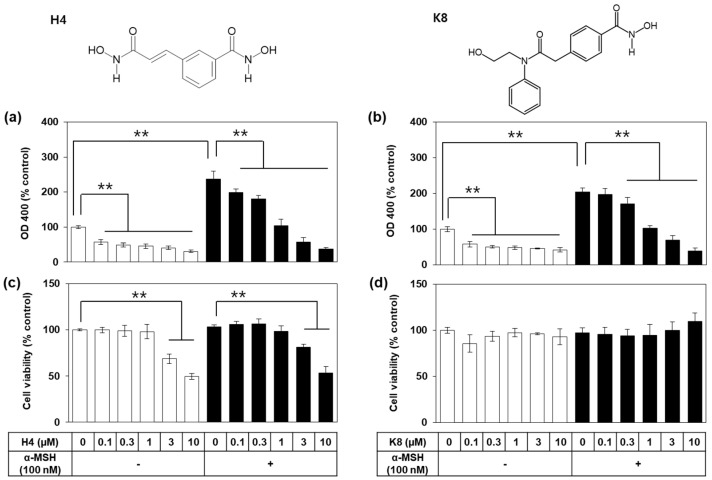
Effects of H4 and K8 on the melanin synthesis and viability of B16-F10 cells. Cells were treated with each drug at various concentrations and stimulated with 100 nM α-melanocyte-stimulating hormone (α-MSH) for 72 h for melanin synthesis assay (**a**,**b**) or for 48 h for relative viability assay (c, d). Data are presented as percentages relative to the control value (mean ± SD, *n* = 4 for a and b; *n* = 3 for (**c**,**d**)). ** *p* < 0.01. Chemical structures of H4 (*N*-hydroxy-3-[3-(hydroxyamino)-3-oxo-1-propen-1-yl-benzamide; CBHA) and K8 (4-((hydroxyamino)carbonyl)-*N*-(2-hydroxyethyl)-*N*-phenyl-benzeneacetamide; HPOB) are shown. Open and closed histograms represent the absence and presence of α-MSH addition, respectively.

**Figure 3 ijms-21-04589-f003:**
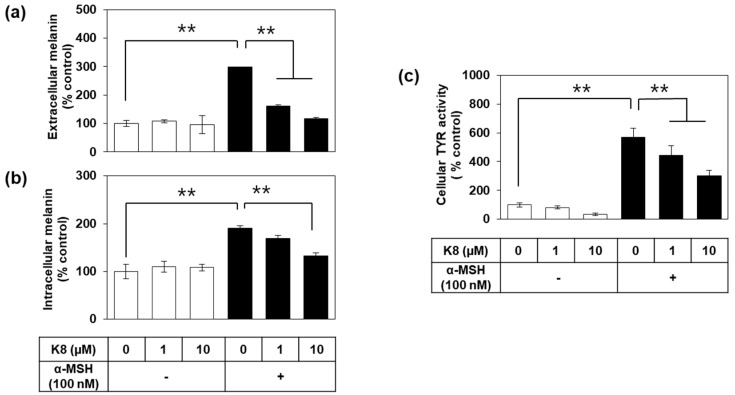
K8 reduces the melanin content and tyrosinase (TYR) activity in B16-F10 cells stimulated with α-melanocyte-stimulating hormone (α-MSH). In (**a**,**b**), cells were stimulated with 100 nM α-MSH for 60 h in the presence or absence of K8 at the indicated concentrations. Extra- and intracellular melanin levels were determined using the conditioned medium and cell extracts, respectively, and the values were normalized to total cellular protein content (*n* = 3). In (**c**), cells were stimulated with 100 nM α-MSH in the presence or absence of K8 at various concentrations for 24 h. Cell lysates were used for the enzyme activity assay (*n* = 4). Data are presented as percentages relative to control value (mean ± SD). ** *p* < 0.01. Open and closed histograms represent the absence and presence of α-MSH addition, respectively.

**Figure 4 ijms-21-04589-f004:**
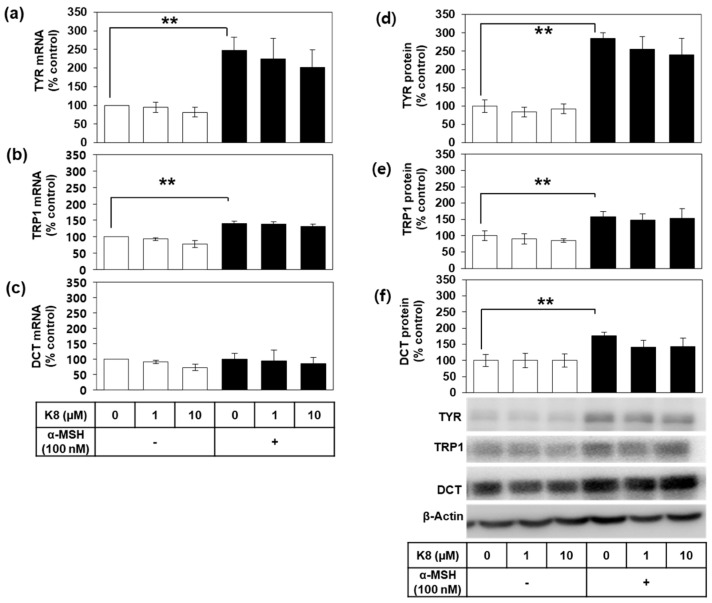
Effects of K8 on the mRNA and protein levels of tyrosinase (TYR), tyrosinase-related protein 1 (TYRP1) and dopachrome tautomerase (DCT) in B16-F10 cells stimulated with alpha-melanocyte-stimulating hormone (α-MSH). Cells were stimulated with 100 nM α-MSH in the presence or absence of K8 at different concentrations for 12 h for mRNA analysis or for 24 h for protein analysis. The mRNA levels of TYR, TYRP1, and DCT were determined by quantitative reverse transcription polymerase chain reaction (qRT-PCR) and normalized to glyceraldehyde 3-phosphate dehydrogenase (GAPDH) (**a**–**c**). Western blotting of cell lysates was performed for TYR, TYRP1, and DCT using β-actin as a loading control (**d**–**f**). Typical blot images are shown. Data are presented as percentages relative to the control value (mean ± SD, *n* = 3 for a, b, and c; *n* = 4 for d, e, and f). ** *p* < 0.01. Open and closed histograms represent the absence and presence of α-MSH addition, respectively.

**Figure 5 ijms-21-04589-f005:**
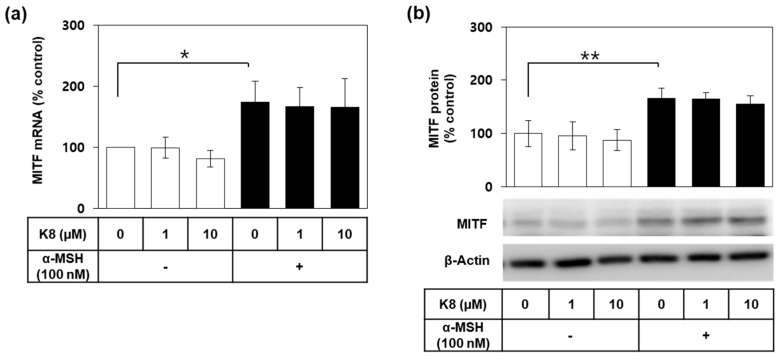
Effects of K8 on the mRNA and protein levels of microphthalmia-associated transcription factor (MITF) in B16-F10 cells stimulated with alpha-melanocyte-stimulating hormone (α-MSH). Cells were stimulated with 100 nM α-MSH in the presence or absence of K8 at various concentrations for 12 h for the mRNA analysis or for 24 h for the protein analysis. The mRNA level of MITF was determined by quantitative reverse transcription polymerase chain reaction (qRT-PCR) and normalized to glyceraldehyde 3-phosphate dehydrogenase (GAPDH) (**a**). The protein level of MITF was determined by Western blotting using β-actin as a loading control (**b**). Typical blot images are shown. Data are presented as percentages relative to the control value (mean ± SD, *n* = 3 for a; *n* = 4 for b). ** *p* < 0.01, * *p* < 0.05. Open and closed histograms represent the absence and presence of α-MSH addition, respectively.

**Figure 6 ijms-21-04589-f006:**
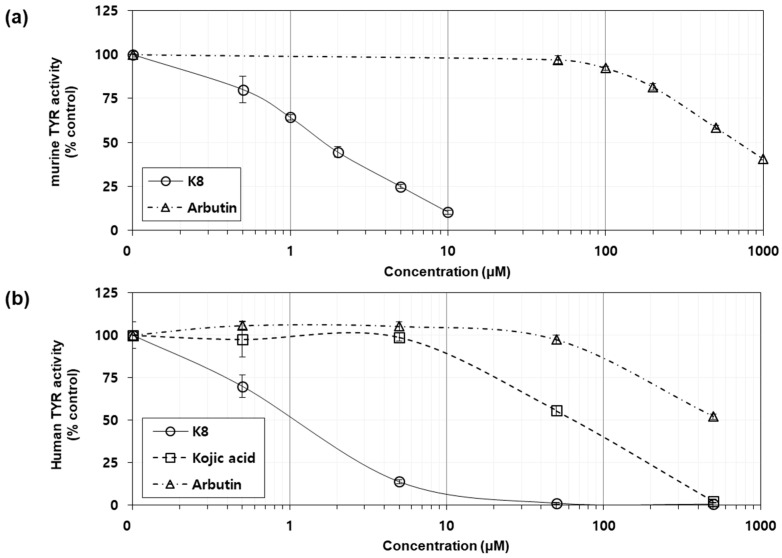
Effects of K8, β-arbutin and kojic acid on the catalytic activity of human tyrosinase (TYR) in vitro. The activity of TYR was determined in the presence of each test compound at various concentrations, using the cell-free extracts of murine B16F10 cells (**a**) and HEK293-TYR cells (**b**). Data are presented as percentages of uninhibited activities (mean ± SD, *n* = 4).

**Figure 7 ijms-21-04589-f007:**
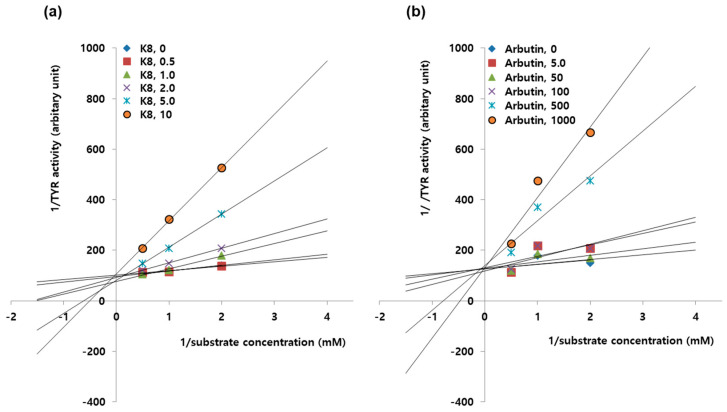
Enzyme kinetics for the inhibition of TYR activity by K8 and β-arbutin. For the enzyme kinetics study of murine TYR, reaction rates were determined at varying concentrations of substrates. Assays were also performed in the presence of K8 (**a**) or β-arbutin (**b**) at the indicated concentrations (μM). Lineweaver–Burk plots are drawn using the mean values of three different measurements in order to determine the inhibition types of K8 and β-arbutin. Both K8 and β-arbutin show competitive inhibition modes against TYR activity.

**Figure 8 ijms-21-04589-f008:**
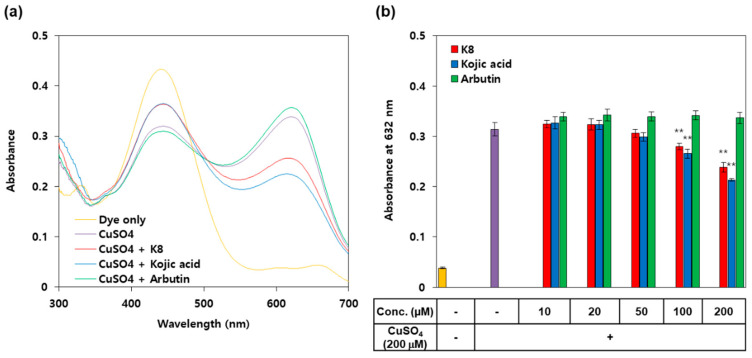
Comparison of copper chelating activities of K8, kojic acid, and β-arbutin. (**a**) Absorption spectra of 200 μM pyrocatechol violet in the absence and presence of 200 μM CuSO_4_ and other additives (200 μM). (**b**) Pyrocatechol violet (200 μM) was reacted with 200 μM CuSO_4_ in the presence of varied concentrations of K8, kojic acid, and β-arbutin. Absorbance of the reaction mixture was determined at 632 nm in order to evaluate copper chelating activities (mean ± SD, *n* = 3). ** *p* < 0.01 versus control.

**Figure 9 ijms-21-04589-f009:**
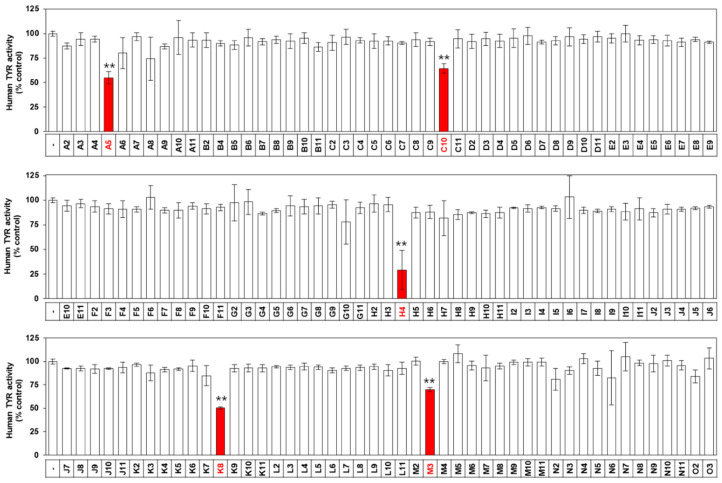
Effects of drugs on human TYR activity in vitro. The activity of human TYR was determined in the absence or presence of each drug (1.0 µM). Data are presented as percentages of uninhibited activity (mean ± SD, *n* = 3). ** *p* < 0.01 vs. vehicle control. Drugs that inhibited the catalytic activity of TYR in vitro are highlighted in red.

**Table 1 ijms-21-04589-t001:** Epigenetic screening library used in this study.

Code	CAS Number	Drug	Code	CAS Number	Drug
A2	21293-29-8	(+)-Abscisic Acid	H4	174664-65-4	CBHA
A3	3544-24-9	3-amino Benzamide	H5	251456-60-7	M 344
A4	929016-96-6	SB939	H6	151720-43-3	Oxamflatin
A5	950762-95-5	PCI 34051	H7	1105698-15-4	Salermide
A6	1219807-87-0	4-iodo-SAHA	H8	299953-00-7	Mirin
A7	410536-97-9	Sirtinol	H9	937039-45-7	Pimelic Diphenylamide 106
A8	328968-36-1	C646	H10	940943-37-3	KD 5170
A9	1239262-52-2	Tubastatin A (trifluoroacetate salt)	H11	404950-80-7	Panobinostat
A10	78824-30-3	Garcinol	I2	209783-80-2	MS-275
A11	476-66-4	Ellagic Acid	I3	926908-04-5	HNHA
B2	287383-59-9	Scriptaid	I4	48208-26-0	RG-108
B4	183506-66-3	Apicidin	I5	10302-78-0	2′,3′,5′-triacetyl-5-Azacytidine
B5	83209-65-8	HC Toxin	I6	979-92-0	S-Adenosyl homocysteine
B6	1238673-32-9	UNC0321 (trifluoroacetate salt)	I7	1197196-48-7	UNC0224
B7	72877-50-0	(−)-Neplanocin A	I8	743420-02-2	Chidamide
B8	1373232-26-8	Cl-Amidine (hydrochloride)	I9	537049-40-4	Tubacin
B9	877617-46-4	F-Amidine (trifluoroacetate salt)	I10	102052-95-9	3-Deazaneplanocin A
B10	1256375-38-8	JGB1741	I11	58944-73-3	Sinefungin
B11	1260635-77-5	coumarin-SAHA	J2	382180-17-8	Pyroxamide
C2	1260907-17-2	I-BET762	J3	5262-39-5	*N*-Oxalylglycine
C3	1255580-76-7	UNC0638	J4	890190-22-4	WDR5-0103
C4	880487-62-7	Phthalazinone pyrazole	J5	1396772-26-1	EPZ005687
C5	961-29-5	Isoliquiritigenin	J6	1561178-17-3	SGC0946
C6	1207113-88-9	CCG-100602	J7	1415800-43-9	UNC1215
C7	1243583-88-1	CAY10669	J8	420831-40-9	AK-7
C8	3690-10-6	Zebularine	J9	1346704-33-3	GSK343
C9	528-53-0	Delphinidin (chloride)	J 10	1619994-69-2	Bromosporine
C10	732302-99-7	ITF 2357	J 11	1619994-68-1	GSK2801
C11	1320288-19-4	UNC0631	K2	14513-15-6	SIRT1/2 Inhibitor IV
D2	1320288-17-2	UNC0646	K3	2147701-33-3	I-CBP112 (hydrochloride)
D3	1310877-95-2	Methylstat (hydrate)	K4	1613695-14-9	SGC-CBP30
D4	120964-45-6	3-Deazaneplanocin A (hydrochloride)	K5	1481677-78-4	UNC0642
D5	129-46-4	Suramin (sodium salt)	K6	1431612-23-5	UNC1999
D6	98-92-0	Nicotinamide	K7	1627607-87-7	(R)-PFI-2 (hydrochloride)
D7	207671-42-9	2,4-Pyridinedicarboxylic Acid	K8	1429651-50-2	HPOB
D8	1403764-72-6	PFI-1	K9	96017-59-3	2-hexyl-4-Pentynoic Acid
D9	320-67-2	5-Azacytidine	K10	1819363-80-8	PFI-3
D10	1020149-73-8	SGI-1027	K11	199596-05-9	JIB-04
D11	2353-33-5	Decitabine	L2	1477949-42-0	CAY10683
E2	1300031-49-5	I-BET151	L3	1346574-57-9	GSK126
E3	1268524-70-4	(+)-JQ1	L4	1446144-04-2	CPI-203
E4	1268524-71-5	(−)-JQ1	L5	154-42-7	6-Thioguanine
E5	160003-66-7	BSI-201	L6	1252003-15-8	Tubastatin A
E6	86-55-5	1-Naphthoic Acid	L7	1968-05-4	3,3′-Diindolylmethane
E7	1716-12-7	Sodium 4-Phenylbutyrate	L8	202590-98-5	OTX015
E8	459868-92-9	Rucaparib (phosphate)	L9	2140-61-6	5-Methylcytidine
E9	5852-78-8	IOX1	L10	304896-21-7	AGK7
E10	1271738-62-5	MI-2 (hydrochloride)	L11	838-07-3	5-Methyl-2′-deoxycytidine
E11	1934302-23-4	MI-nc (hydrochloride)	M2	1380288-87-8	EPZ5676
F2	95058-81-4	Gemcitabine	M3	852475-26-4	MC 1568
F3	192441-08-0	Lomeguatrib	M4	40951-21-1	α-Hydroxyglutaric Acid (sodium salt)
F4	1216744-19-2	GSK4112	M5	52248-03-0	S-(5′-Adenosyl)-L-methionine (tosylate)
F5	876150-14-0	Octyl-α-ketoglutarate	M6	1044870-39-4	RVX-208
F6	1596-84-5	Daminozide	M7	1012054-59-9	CUDC-101
F7	1797832-71-3	GSK-J1 (sodium salt)	M8	404951-53-7	LAQ824
F8	2108665-15-0	GSK-J2 (sodium salt)	M9	300816-11-9	Nullscript
F9	1797983-09-5	GSK-J4 (hydrochloride)	M10	1431368-48-7	GSK-LSD1 (hydrochloride)
F10	112522-64-2	CI-994	M11	1357389-11-7	RGFP966
F11	2108899-91-6	CPTH2 (hydrochloride)	N2	1440209-96-0	BRD73954
G2	33419-42-0	Etoposide	N3	501-36-0	*trans*-Resveratrol
G3	111358-88-4	Lestaurtinib	N4	89464-63-1	DMOG
G4	778649-18-6	Butyrolactone 3	N5	58880-19-6	Trichostatin A
G5	1069-66-5	Valproic Acid (sodium salt)	N6	193551-00-7	CAY10398
G6	380315-80-0	Tenovin-1	N7	1418131-46-0	RSC-133
G7	1011557-82-6	Tenovin-6	N8	537034-17-6	BML-210
G8	156-54-7	Sodium Butyrate	N9	10083-24-6	Piceatannol
G9	1808255-64-2	BIX01294 (hydrochloride hydrate)	N10	839699-72-8	CAY10591
G10	16611-84-0	Anacardic Acid	N11	848193-68-0	EX-527
G11	304896-28-4	AGK2	O2	149647-78-9	SAHA
H2	1045792-66-2	CAY10603	O3	1986-47-6	2-PCPA (hydrochloride)
H3	5690-03-9	Splitomicin

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
