# Peer review of "Screening of an Epigenetic Drug Library Identifies 4-((hydroxyamino)carbonyl)-N-(2-hydroxyethyl)-N-Phenyl-Benzeneacetamide that Reduces Melanin Synthesis by Inhibiting Tyrosinase Activity Independently of Epigenetic Mechanisms"

_ijms, 2020, doi:10.3390/ijms21134589_

Round 1

Reviewer 1 Report

This manuscript is interesting and nicely written. The narrative can easily be followed. I only have a few small comments or suggestions:

Line 15: aim of this study was to identify (use past tense)

Line 23: be specific and mention if alpha- or beta-arbutin was used. Applies for entire manuscript.

Line 24: copper chelating activity?

Line 26: through copper

Line 48: Omit ‘between races’. There is no such thing as racial or ethnical skin colour determination.

Line 97: please provide the ATCC number for the B16-F10 cell line.

Section 2.3.: How were the cells lysed?

There is a difference in the amount of cells plated in section 2.3 vs 2.5. Why is this?

Line 169: Image J program?

Line 241: There is a small typo in the Figure caption: 0.01≤ should be removed.

Line 243: same as Line 241

Figure 5: There is a # shown in the graphs, but a ** in the caption. It should also mention versus what it is significant.

Figure 6: again, please mention which arbutin (alpha or beta) was used.

Figure 7: same as in figure 6 for arbutin.

Line 373: Sentence is either too much or missing a part.

Line 382: I am not sure if one can say K8 has good safety. It is not cytotoxic at the tested concentrations, but the study does not describe safety tests as such, like e.g. sensitization or mutagenicity tests. I think this statement should be removed or safety tests should be described.

Line 226-228: The authors mention that unexpectedly 10 uM K8 only partially lowered TYR activity while suppressing almost all melanin synthesis. Why was this unexpected, and what is your hypothesis for this finding? Please add a short comment to the Discussion.

Author Response

Reviewer 1.

This manuscript is interesting and nicely written. The narrative can easily be followed. I only have a few small comments or suggestions:

Line 15: aim of this study was to identify (use past tense)

: Corrected. (Line 15)

Line 23: be specific and mention if alpha- or beta-arbutin was used. Applies for entire manuscript.

: It is beta form. Arbutin was changed to beta-arbutin throughout the paper. (Line 23…)

Line 24: copper chelating activity?

: Corrected. (Line 24)

Line 26: through copper

: Corrected. (Line 26)

Line 48: Omit ‘between races’. There is no such thing as racial or ethnical skin colour determination.

: Deleted as suggested. Thanks. (Line 48)

Line 97: please provide the ATCC number for the B16-F10 cell line.

: Added as suggested. (Line 97)

Section 2.3.: How were the cells lysed?

: OD was measured without cell lysis. It was indicated (Line 105).

There is a difference in the amount of cells plated in section 2.3 vs 2.5. Why is this?

: Because the cell growth areas of each well in a 96-well plate and 6-well plates are different.

Line 169: Image J program?

: Corrected. (Line 170)

Line 241: There is a small typo in the Figure caption: 0.01≤ should be removed.

: Corrected. (Line 245)

Line 243: same as Line 241

: Corrected. (Line 247)

Figure 5: There is a # shown in the graphs, but a ** in the caption. It should also mention versus what it is significant.

: The symbols in the Figure 5 were corrected.

Figure 6: again, please mention which arbutin (alpha or beta) was used.

: Arbutin was changed to beta-arbutin. (Line 307)

Figure 7: same as in figure 6 for arbutin.

: Arbutin was changed to beta-arbutin. (Line 318)

Line 373: Sentence is either too much or missing a part.

: The ghost sentence was deleted. (Line 383)

Line 382: I am not sure if one can say K8 has good safety. It is not cytotoxic at the tested concentrations, but the study does not describe safety tests as such, like e.g. sensitization or mutagenicity tests. I think this statement should be removed or safety tests should be described.

: “and safety” was deleted as suggested. (Line 394)

Line 226-228: The authors mention that unexpectedly 10 uM K8 only partially lowered TYR activity while suppressing almost all melanin synthesis. Why was this unexpected, and what is your hypothesis for this finding? Please add a short comment to the Discussion.

: We added the following statement in Discussion. (Line 389)

Our original assumption was that K8 might inhibit melanin synthesis by suppressing TYR expression to the same extent (as monitored by Western blotting and enzyme assay using the cell lysates) but it turned out not to be the case.

Reviewer 2 Report

The authors report a study aimed at finding a new depigmenting agent acting via an epigenetic mechanism, using an epigenetic library. Finally the best candidate tested, K8, acts via a non-epigenetic mechanism.

The strategy is interesting and the methods well adapted for an antimelanogenic activity in vitro.

Major comments :

  1. According to the Figure 1, several compounds are more potent than K8 as inhibitors of melanogenesis. They have been discarded because of their cytotoxicity at 1 µM. However, some of these compounds may have a therapeutic window if they inhibit melanin synthesis at lower concentrations than those inducing cytotoxicity. It would have been interesting to compare their IC50on melanin synthesis and MTT assay (cytotoxicity). Which is important for a clinical application is not the absolute concentration (dose) for an adverse event (cytotoxicity), but the ratio between the effective dose for the therapeutic effect and that for adverses events. Doing so, the authors may have missed interesting candidates for a new depigmenting agent.
  2. Many compounds have been shown to have an antimelanogenic activity targeting tyrosinase in vitro, however almost none of them exert a depigmenting activity in vivo. This should call into question the use of tyrosinase to screen for depimenting agents. One problem may the bioavailability to melanocytes of compounds applied on the skin; another one could be the fact that it is not sufficient to inhibit new melanin synthesis for a depigmenting activity, but the degradation of existing melanin is required too. This could be added in the discussion.

Minor comments :

  1. Methods, cytotoxicity : you may use the common abbreviation MTT for 3-(4,5-dimethylthiazol-2-yl)-2,5-diphenyltetrazolium bromide.
  2. Methods, Western blots : please indicate the concentrations/dilutions used for the primary and secondary antibodies.
  3. Methods, copper chelation. Please briefly indicate the principle of the methods : the competition between the chelating properties of pyrocatechol violet and the tested chelating agents decreases the absorption at 632 nm, which is the maximum for the [copper-pyrocatechol violet] complex.
  4. Figure 2 legend : please indicate the difference between open and closed histogrammes (it is only mentioned in the text).

Author Response

Reviewer 2.

Major comments:

  1. According to the Figure 1, several compounds are more potent than K8 as inhibitors of melanogenesis. They have been discarded because of their cytotoxicity at 1 µM. However, some of these compounds may have a therapeutic window if they inhibit melanin synthesis at lower concentrations than those inducing cytotoxicity. It would have been interesting to compare their IC50on melanin synthesis and MTT assay (cytotoxicity). Which is important for a clinical application is not the absolute concentration (dose) for an adverse event (cytotoxicity), but the ratio between the effective dose for the therapeutic effect and that for adverse events. Doing so, the authors may have missed interesting candidates for a new depigmenting agent.

: We agree, and the following statement is added to Discussion: (Line 406)

The drugs that had showed cytotoxicity at 1 µM (Figure 1) were eliminated from the active drug candidates. The eliminated drugs might have inhibited melanin synthesis without cytotoxicity if they had been assayed at lower concentrations. Therefore further studies are needed to examine this possibility. Because only 5 out of 141 drugs inhibited the catalytic activity of TYR (Figure 1), additional active drugs, if found any, are expected to inhibit melanin synthesis by mechanisms other than inhibition of TYR catalytic activity.

  1. Many compounds have been shown to have an antimelanogenic activity targeting tyrosinase in vitro, however almost none of them exert a depigmenting activity in vivo. This should call into question the use of tyrosinase to screen for depimenting agents. One problem may the bioavailability to melanocytes of compounds applied on the skin; another one could be the fact that it is not sufficient to inhibit new melanin synthesis for a depigmenting activity, but the degradation of existing melanin is required too. This could be added in the discussion.

: We agree that TYR inhibitors may be very difficult to show a depigmenting effect, but we do not think it is impossible. The following statement is added to Discussion: (Line 444)

The skin pigmentation degree will be determined by the relative balance between the pigmentation processes, such as melanin synthesis, and the depigmentation processes, such as elimination of stratum corneum. We have previously reported that topically applied resveratrol derivatives, which inhibited melanin synthesis through multiple mechanisms including inhibition of TYR catalytic activity [56, 58, 64], showed depigmentation efficacy in human trials [46, 47]. Thus, if the depigmentation process is in the same condition, inhibition of melanin synthesis may brighten the skin tone, and a potent TYR inhibitor may be a candidate drug. Further research is needed to investigate whether the active drug found in this study can be delivered to melanocytes well and whether it can actually exhibit pigmentation efficacy in vivo.

Minor comments:

3. Methods, cytotoxicity: you may use the common abbreviation MTT for 3-(4,5-dimethylthiazol-2-yl)-2,5-diphenyltetrazolium bromide.

: Used abbreviation MTT as suggested. (Line 109)

4. Methods, Western blots: please indicate the concentrations/dilutions used for the primary and secondary antibodies.

: Described as suggested. (Line 157)

Each antibody was diluted 1 to 1000 in antibody dilution buffer containing 20 mM Tris-Cl (pH 7.5), 200 mM NaCl and 5% skim milk.

5. Methods, copper chelation. Please briefly indicate the principle of the methods: the competition between the chelating properties of pyrocatechol violet and the tested chelating agents decreases the absorption at 632 nm, which is the maximum for the [copper-pyrocatechol violet] complex.

: The following statement is added in Methods. (Line 190)

The principle of this method is that, when the pyrocatechol violet is complexed with copper, it shows maximum absorption at 632 nm, and the test material competing with pyrocatechol violet for copper can reduce the absorbance of the [copper-pyrocatechol violet] complex.

6. Figure 2 legend: please indicate the difference between open and closed histograms (it is only mentioned in the text)

: The following statement is added in Figures 1, 2, 3, 4, and 5 legends. (Line 257)

Open and closed histograms represent the absence and presence of alpha-MSH addition, respectively.